# Chinese health funding in Africa: The untold story

**Carrie B. Dolan**[1,2,3]*, **Ammar A. Malik**[3,4], **Sheng Zhang**[3,4], **Wenhui Mao**[5], **Kaci Kennedy McDade**[5], **Eli Svoboda**[2], **Julius N. Odhiambo**[2,3]

**1** Department of Health Sciences, William & Mary, Williamsburg, Virginia, United States of America, **2** Ignite Global Health Research Lab, William & Mary, Williamsburg, Virginia, United States of America, **3** Global Research Institute, William & Mary, Williamsburg, Virginia, United States of America, **4** AidData, William & Mary, Williamsburg, Virginia, United States of America, **5** Center for Policy Impact in Global Health, Duke University, Durham, NC, United States of America

* cbdolan@wm.edu

**Data Availability Statement:** Data are available in a public, open access repository. Data are available from: https://www.aiddata.org/publications/aiddata-tuff-methodology-version-2-0.

## Abstract

The motivations behind China's allocation of health aid to Africa remain complex due to limited information on the details of health aid project activities. Insufficient knowledge about the purpose of China's health aid hinders our understanding of China's comprehensive role in supporting Africa's healthcare system. To address this gap, our study aimed to gain better insights into China's health aid priorities and the factors driving these priorities across Africa. To achieve this, we utilized AidData's Chinese Official Finance Dataset and adhered to the Organisation for Economic Co-operation and Development (OECD) guidelines. We reclassified all 1,026 health projects in Africa, originally categorized under broad 3-digit OECD-DAC sector codes, into more specific 5-digit CRS codes. By analyzing the project count and financial value, we assessed the shifting priorities over time. Our analysis revealed that China's priorities in health aid have evolved between 2000 and 2017. In the early 2000s, China primarily allocated aid to basic health personnel and lacked diversity in sub-sectors. However, after 2004, China shifted its focus more toward basic infrastructure and reduced emphasis on clinical-level staff. Furthermore, China's interest in addressing malaria expanded both in scale and depth between 2006 and 2009. This trend continued in 2012 and 2014 when China responded to the Ebola outbreak by shifting its focus from basic infrastructure to infectious diseases. In summary, our findings demonstrate the changes in China's health aid strategy, starting with addressing diseases already eliminated in China and gradually transitioning towards global health security, health system strengthening, and shaping the governance mechanisms.

## Introduction

China invested over $5.6 billion in health funding worldwide between 2000 and 2017, including $1.6 billion in Africa alone [1]. During this time, China has emerged as a prominent donor devoted to improving health systems through investments in physical infrastructure such as

**Funding:** The author(s) received no specific funding for this work

**Competing interests:** The authors have no competing interests to declare

hospitals, provision of critical equipment such as CT Scanners, and technical support targeted at medical workers [2, 3]. China has a long history of in-country health system reform. Since the 1950s, improvements in China's healthcare system have contributed substantially to reducing the burden of infectious, maternal, and neonatal diseases at home [4]. A substantial investment in health systems, relevant recent experience, and recent economic development make China distinctive as a donor and strategic partner positioned to strengthen African health systems.

Evidence suggests four main components of China's health aid to Africa: the placement of Chinese Medical Teams; donations of medical supplies and equipment; building health infrastructure in the form of construction of clinics and hospitals; and public health program support in the forms of malaria and Ebola treatment as well as training programs [5]. However, understanding the motivations for Chinese health sector funding remains complicated. China's most recent official white paper on foreign aid portrays Chinese aid as motivated by "pursuing the greater good and shared interests, with higher priority given to the former" [6]. Some evidence supports the notion that Chinese health aid is genuinely altruistic, indicating that China, to some extent, is concerned with other country's needs [7–9]. For instance, China targets more health aid overall to poorer countries [7, 10]. The most common perspective is that China provides health aid to elevate its international image or to protect national security by reducing global pandemics that cause domestic disease outbreaks and economic decline [9–15]. Most evidence supporting this belief is qualitative. Only one relevant quantitative study was found. Yang et al. [10] used the frequency of health aid programs in capital and politician-birth cities to indicate China's attempts to improve its international image.

Another popular viewpoint is that China intends to secure access to recipients' valuable natural resources in return for health aid [9, 13–16]. However, three recent studies found no significant relationship between natural resources and China's health aid allocation [7, 8, 17]. Other scholars propose that China hopes to expand exports and improve terms of trade with recipient countries [14, 15, 18]. Investigating the relationship between Chinese health aid and trade interests has yielded mixed results [2, 19]. Furthermore, some evidence indicates that China uses health aid to promote specific political interests, particularly to call attention to its opposition to Taiwan's political independence [7, 12, 13, 18]. For example, countries with positive diplomatic relationships with Taiwan are almost entirely excluded from Chinese health aid [7]. Additionally, Yang et al.'s study [2018] implies that China allocates most of its health aid projects in Africa along the continent's western and eastern coasts and that China is disproportionately likely to allocate projects to capital cities with large population densities [10]. The authors suspect China is particularly concerned with assisting the recipient elite class rather than the overall population. Refuting claims of an intentional allocation strategy, Shen & Fan [2014] contend that literature typically wrongfully assumes that China follows a unified plan regarding health aid. Rather, Shen & Fan argue that the varying motives of China's provinces result in a decentralized, unpredictable health aid policy that consists of distinct and often conflicting parts [20].

The motivations behind China's health aid allocation remain complicated because the details around health aid project activities remain unclear. We know too little about the purpose of China's health aid in Africa, and we believe this incomplete information prohibits understanding China's full role in supporting Africa's healthcare system. Moreover, China abstains from systematically reporting its foreign development assistance activities to established international channels, such as the OECD's Common Reporting System [OECD-CRS] or the International Aid Transparency Initiative [IATA]; therefore, its increasingly salient role in health remains understudied. While recent studies investigate the determinants of Chinese health aid allocation, these studies use earlier data only covering commitment years 2000–2014. This dataset was significantly less granular regarding spatial, temporal, and technical

details. With only 620 Chinese-financed healthcare projects covered in the entire dataset, these studies provide a limited understanding of the true scope of China's global health interventions, particularly in Africa. Following the Chinese health aid allocation literature, we narrow the evidence gap through the most up-to-date look at granular data on priorities and changes to China's health aid portfolio over time. We first provide overarching information about a newly available dataset [AidData's Global Chinese Development Finance Dataset Version-2.0 [GCDF 2.0]] [1], including the number of projects and geographic reach. Second, we dive deeper into the changing priorities over time and what those priorities signal. Our goal is exploratory to understand better Chinese health aid priorities and the potential drivers of these priorities. This approach allows us to understand what the health aid flows were designed to address, deserving attention from researchers and policymakers alike.

## Methods

### Data sources

We utilized AidData's Global Chinese Development Finance Dataset Version 2.0 [GCDF 2.0] to fill the information void created by China's non-participation in any international aid transparency initiative. AidData is a non-profit research institution based at William & Mary [1], GCDF 2.0 offers unprecedented detail on 1,448 Chinese-financed healthcare interventions worldwide between 2000–2017, including 1,026 in Africa, worth $1.6 billion [1]. Since 2013, AidData has maintained the world's most comprehensive dataset, tracking granular details on China's overseas development financing program. This dataset is publicly available from AidData and is assembled using the TUFF [Tracking Underreported Financial Flows] methodology designed to capture the complete set of international official development financing flows. This is done by organizing extensive official source catalogs for each country, backed by media sources, from where trained staff create project records which later undergo multi-stage data quality reviews. The dataset explicitly focuses on donor countries not participating in established global aid transparency initiatives. This data includes all official sector financial flows from China that align with OECD criteria for ODA and OOF, including grants, loans, export credits, technical assistance, debt forgiveness, and rescheduling. In addition, it captures projects over eighteen financial commitment years [2000–2017], with details on the timing of project implementation over 22 years [2000–2021].

The project subsets the GCDF 2.0 data to only include health and population policies/programs and reproductive health sector projects [CRS activity codes 120 and 130] in Africa. Not all projects in the GCDF 2.0 data were assigned commitment values. We include all projects regardless of commitment value, but projects with missing financial data were ignored in the commitment calculations. Therefore, the dollar estimates utilized in this analysis must be interpreted cautiously. Only formally approved, active, and completed projects were included in the analysis. The data are categorized as Official Development Assistance [ODA]-like, Other Official Flows [OOF]-like, or Vague [official Finance] based on Organisation for Economic Co-operation and Development Assistance Committee [OECD-DAC] guidelines for ODA and OOF. This analysis only included ODA-like projects, resulting in projects with development intent that were concessional and took place in a country that qualifies for ODA based on income level. Only 26 projects in GCDF, or 2.4% of all health projects in Africa, fall into the OOF and vague categories, nearly all of which are export buyer's credits for medical equipment purchases. Based on the GCDF 2.0 and OECD-DAC guidelines, we built on the GCDF 2.0 data and designed a detailed protocol for reclassifying all 1,026 health projects in Africa from their broad 3-digit OECD-DAC sector codes to their more specific 5-digit CRS codes [1, 21]. We did this by following OECD's sector classification guidelines, which provide

guidelines and detailed nesting information of 5-digit codes within each of the 24 sector categories represented at the 3-digit code level. For example, whereas all health sector [code: 120] projects are 3-digit tagged, activities most closely aligned with "tuberculosis control" [code: 12263] and "basic health infrastructure" [code: 12230] are allocated their respective 5-digit codes. In our analysis, we included the categories of "basic care" and "medical services," which are similar but not identical. Basic care directly delivers health services and trains local staff, whereas "medical services" refers primarily to medical equipment and staffing. Projects involving activities like "nursing home" does not fit into basic health since it is specialized and therefore were coded as medical services. We followed a double-blind review process for coding health projects into 5-digit codes for sub-sectors, i.e., two independent coders reviewed each project to ensure internal consistency. In cases with discrepancies, plus an additional 10% of the sample picked randomly, senior coders undertook additional quality assurance processes by reviewing coder determinations for consistency and accuracy. To compare China to more traditional global health donors, project-level data from CRS [purpose codes 12110, 12181, 12182, 12191, 12220, 12230, 12240, 12250, 12261, 12262, 12263, and 12281] were also obtained for the years 2000–2017 from the OECD web portal [accessed April 2022].

## Role of the funding source

This article was not supported by a separate contractor or funding agency but was prepared under the author's sole and independent support.

## Results

### The role of health in China's aid portfolio

As shown in Fig 1, health projects are a small proportion of China's massive global development financing footprint. For example, between 2000–2017, the global health footprint of Chinese commitments was $5.6 billion for 1,448 projects worldwide and $1.6 billion for 1,026 projects in Africa. On average, African health projects have smaller financial commitment sizes. For example, before the 2013 announcement of China's flagship Belt and Road Initiative [BRI], the annual average commitment level was 67 health projects worth $253 million worldwide—representing just 0.8% of the total Chinese development finance portfolio. Nevertheless, in the first five years of the BRI era between 2013 and 2017, when China has generally increased its global project footprint with a focus on connective infrastructure, average health project numbers and dollar values of commitments have doubled, up to 115 and $469 million, respectively. As a result, during the BRI era, China's overseas development financing portfolio value lowered to 0.5%, compared to 0.8% in the pre-BRI period.

### Geographic focus

Whereas Fig 1 focuses on project counts, Fig 2 shows both the intensity of financing commitments and new projects, where darker shades indicate the greater intensity of commitment levels at the country level. Fig 2 was created in ArcGIS 10.7.1 using basemaps from geoBoundaries [22, 23]. Our findings contrast with previous studies on Chinese health investments in Africa, which suggested that resource-rich countries were not given priority. Instead, we found that the countries with the most commitments were Kenya, Zambia, Mauritania, and the Republic of Congo, which are all rich in natural resources.

**Chinese health aid by sub-sector.** Table 1 shows the sub-sectoral breakdown of all African Chinese health aid projects, project counts, commitment, and commitment values. We identified 1,026 health projects allocated to 51 African countries, representing $1.6 billion in

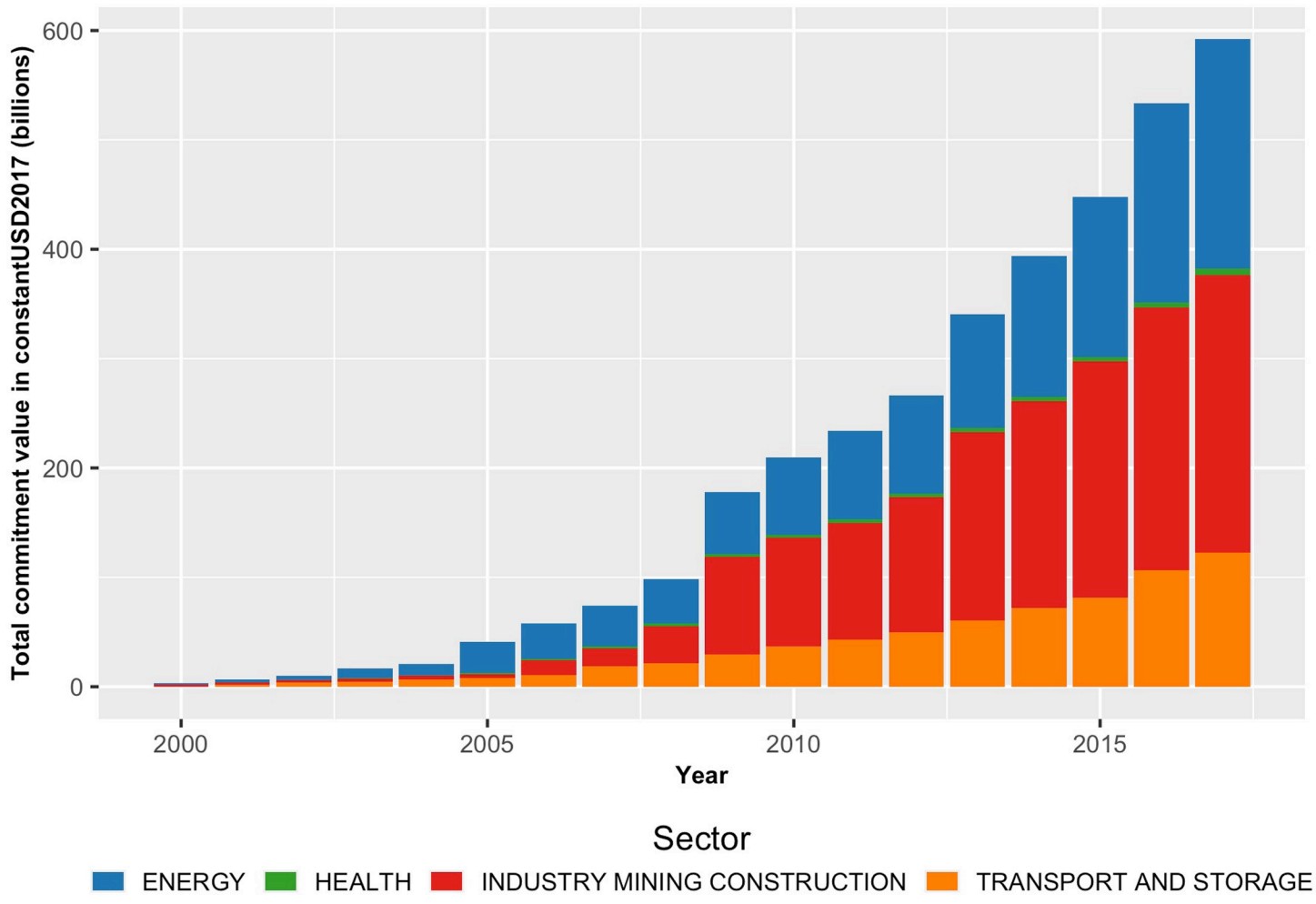

**Fig 1. China's overseas development footprint by sector.**

health aid. The majority of China's projects were for basic health care [49%] and supplied drugs, medicines, and vaccines, followed by malaria control [21%] and basic health infrastructure [18%]. In addition to the data presented in Table 1, we find that country-level total projects vary significantly across the continent, ranging from 2 projects in Egypt and South Africa to 37 in Sierra Leone, 39 in Uganda, and 41 in Tanzania. In terms of temporal trends of project counts, we find that after fluctuating between 17–30 new commitments per year until 2004, by 2009, new Chinese commitments in the continent were topping 89. Later, after stabilizing at around 71 until 2016, commitments crossed the 90 level in 2017. As previously mentioned, not all projects in the GCDF 2.0 data had commitment values; therefore, these estimates should be interpreted cautiously. For example, no medical education/training, population policy, or tuberculosis projects were valued, while 74% of basic health infrastructure projects were assigned a commitment value.

### Shifting priorities over time: China as a demand-responsive donor

China's priorities have shifted over time. Fig 3 shows that during the early 2000s, China allocated aid primarily to basic health personnel [82%] and lacked sub-sector diversity. After 2004 however, China focused more on basic infrastructure and less on clinical-level staff. Their

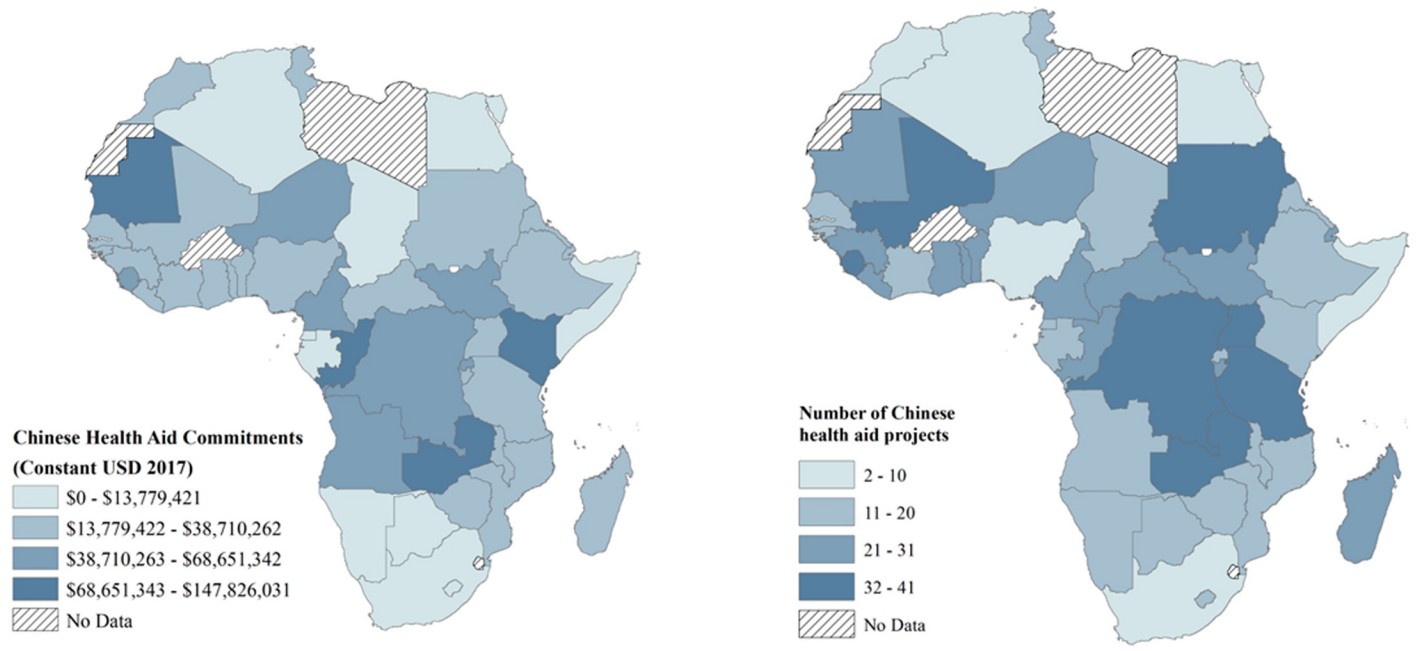

Created by Carrie B. Dolan using ArcGIS 10.7.1
Basemap shapefile accessed from geoBoundaries 4/26/2023
https://www.geoboundaries.org/downloadCGAZ.html
License information for basemap

**Fig 2. Chinese health aid commitments and project counts [2000–2017]. Links:** https://www.geoboundaries.org/downloadCGAZ.html. https://github.com/wmgeolab/geoBoundaries/blob/main/LICENSE.

interest in malaria also expanded in scale and depth between 2006–2009, indicating a sustained commitment to Chinese malaria control projects in Africa. Of the 219 malaria projects in the GCDF 2.0 data, the majority [77%] were for basic health care, mainly in anti-malarial medications. This was followed by aid for basic health infrastructure [18%] that included constructing anti-malarial centers and providing medical equipment. We continued to see this demand response in 2012 and 2014 when China shifted from basic infrastructure to infectious disease in response to Ebola.

In contrast to Fig 3, which focuses on China, Fig 4 focuses on examining the allocation of health aid among the 30 DAC members across all African countries over the same 18-month period. The primary purpose of Fig 4 is to explore the distribution of health aid among the DAC members. This figure reveals a much steadier picture of sectoral allocations. Findings indicate that DAC donors prioritize the same sub-sectors over time, whereas China's focus sub-sectors are evolving. This observation is partly due to the law of averages, i.e., more players and more significant sizes of outlay are naturally more stable than a single donor country's smaller-scale operation. Both China and the OECD health aid projects have maintained a steady trend in their total allocations since 2009 when total allocations rose from under $2 billion to approximately $2.5 billion annually.

## Discussion

China's global health engagement has historically been unique, but an understanding of its role within Africa's healthcare system is emerging [24]. Project details, such as sub-sectors, locations, and durations, have remained mysterious. This lack of transparency is a suboptimal

**Table 1. Summary of health projects in the GCDF 2.0 data.**

| Purpose code category | CRS code | Number of projects | Percentage of total Chinese health aid | Mean commitment value (Constant 2017 US$) | Total commitment value (Constant 2017 US$) | Number of projects with commitment data | Percentage of projects with commitment data |
|---|---|---|---|---|---|---|---|
| Health policy and administrative management | 12110 | 7 | 0.68% | $821,847.43 | $4,109,237.13 | 5 | 71.43% |
| Medical education/training | 12181 | 1 | 0.10% | | | | |
| Medical services | 12191 | 43 | 4.19% | $2,913,822.58 | $46,621,161.28 | 16 | 37.21% |
| Basic health care | 12220 | 504 | 49.12% | $814,214.49 | $60,251,872.13 | 74 | 14.68% |
| Basic health infrastructure | 12230 | 187 | 18.23% | $10,292,416.19 | $1,420,353,433.87 | 138 | 73.80% |
| Infectious disease control | 12250 | 39 | 3.80% | $602,208.45 | $14,453,002.69 | 24 | 61.54% |
| Malaria control | 12262 | 219 | 21.35% | $682,351.23 | $66,188,068.90 | 97 | 44.29% |
| Tuberculosis control | 12263 | 1 | 0.10% | | | | |
| Health personnel development | 12281 | 15 | 1.46% | $11,189,190.09 | $33,567,570.28 | 3 | 20.00% |
| Other prevention and treatment of noncommunicable diseases | 12350 | 1 | 0.10% | $462,745.85 | $462,745.85 | 1 | 100.00% |
| Population policy and administrative management | 13010 | 1 | 0.10% | | | | |
| Reproductive health care | 13020 | 3 | 0.29% | $490,796.81 | $1,472,390.43 | 3 | 100.00% |
| Family planning | 13030 | 2 | 0.19% | $48,662.73 | $97,325.46 | 2 | 100.00% |
| STD control including HIV/AIDS | 13040 | 3 | 0.29% | $214,135.79 | $428,271.58 | 2 | 66.67% |
| TOTAL | | 1026 | 100.00% | | $1,648,005,079.60 | 365 | 35.58% |

outcome on many counts. Health researchers cannot ascertain the true drivers of positive and negative changes in health outcomes. Furthermore, multilateral institutions [e.g., WHO] and bilateral aid agencies [e.g., USAID] cannot coordinate interventions, and recipient governments lack the complete picture of what works and does not work in improving their population's health. Therefore, filling this evidence gap through rigorously collected data on China's financing of Africa's health system is essential to improve public decision-making that would, in turn, enhance public welfare outcomes. Despite the smaller scale of the health sector relative to other sectors, its impact on human lives is more significant per dollar spent because it represents a quintessential public service to citizens.

Our goal was exploratory with the purpose of better understanding China's health aid priorities and potential drivers of these priorities across Africa. Our analysis highlights critical areas for scholars and policymakers to consider. First, China is a global leader in malaria control, successfully eliminating malaria in 2021. The massive Chinese commitment level increase for malaria from 2006 until the Ebola crisis is remarkable. Although the United States is the largest donor government to global malaria efforts, China's commitments are robust. China increased malaria control commitments dramatically [1946%] between 2005 and 2006, likely due to the prioritization of malaria at the Forum on China-Africa Cooperation [FOCAC], setting the stage for regional China-Africa cooperation [25]. Malaria control and eventual eradication were highly successful in China, and much of its outward support resulted from its domestic success [26]. Building on this success, the China-Africa malaria cooperation has expanded in scale and depth. This expansion increased malaria-related funding, including Chinese support for 30 health facilities designed to strengthen Africa's response to malaria through increased access to diagnosis and treatment. These centers have served as a platform to facilitate capacity building by improving local medical research capacity [27]. Physical

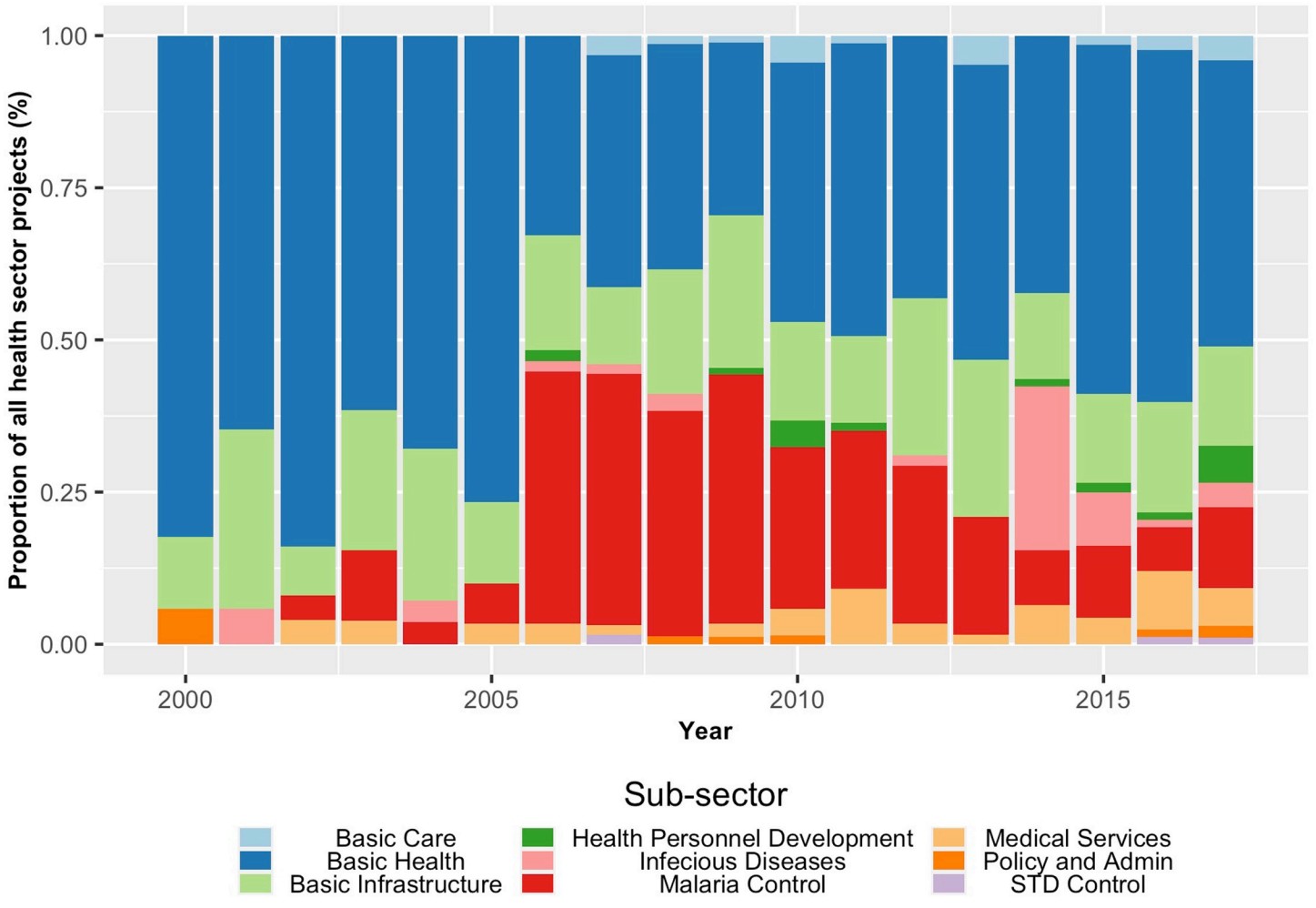

**Fig 3. Primary purpose of Chinese health aid projects in Africa by year [2000–2017; N = 1026].**

infrastructure is one thing, but there are clear examples of capacity building and drug donations that go along with establishing these centers [27]. As evidenced by Xia [2014], "China provided academic exchanges and training programs for African officials and technical personnel. The National Institute of Parasitic Diseases [NIPD] of the Chinese Center for Disease Control and Prevention [CDC] launched five training workshops over the last three years such as 'Infectious diseases prevention and control' and 'prevention and control of malaria and schistosomiasis in developing countries. More than 150 technical staff members and officials from over 20 African countries have been trained in PR China in strategies and measures for malaria prevention and control" [27]. China discovered and now plays a significant role in supplying artemisinin-based combination therapies, now standard WHO treatment recommendations for malaria [26]. Along the same lines of innovation, China is a substantial supplier of long-lasting insecticidal nets, which averted an estimated 68 percent of malaria cases in Africa between 2000 to 2015 [26, 28].

Second, at a descriptive level, China appears highly and rapidly responsive to demand. The most unambiguous indication of this reality came in the aftermath of the Ebola outbreak in 2014. This outbreak caused China's infectious disease spending across the continent to rise 150% between 2013 and 2014, with much of the increase [49%] being allocated to countries

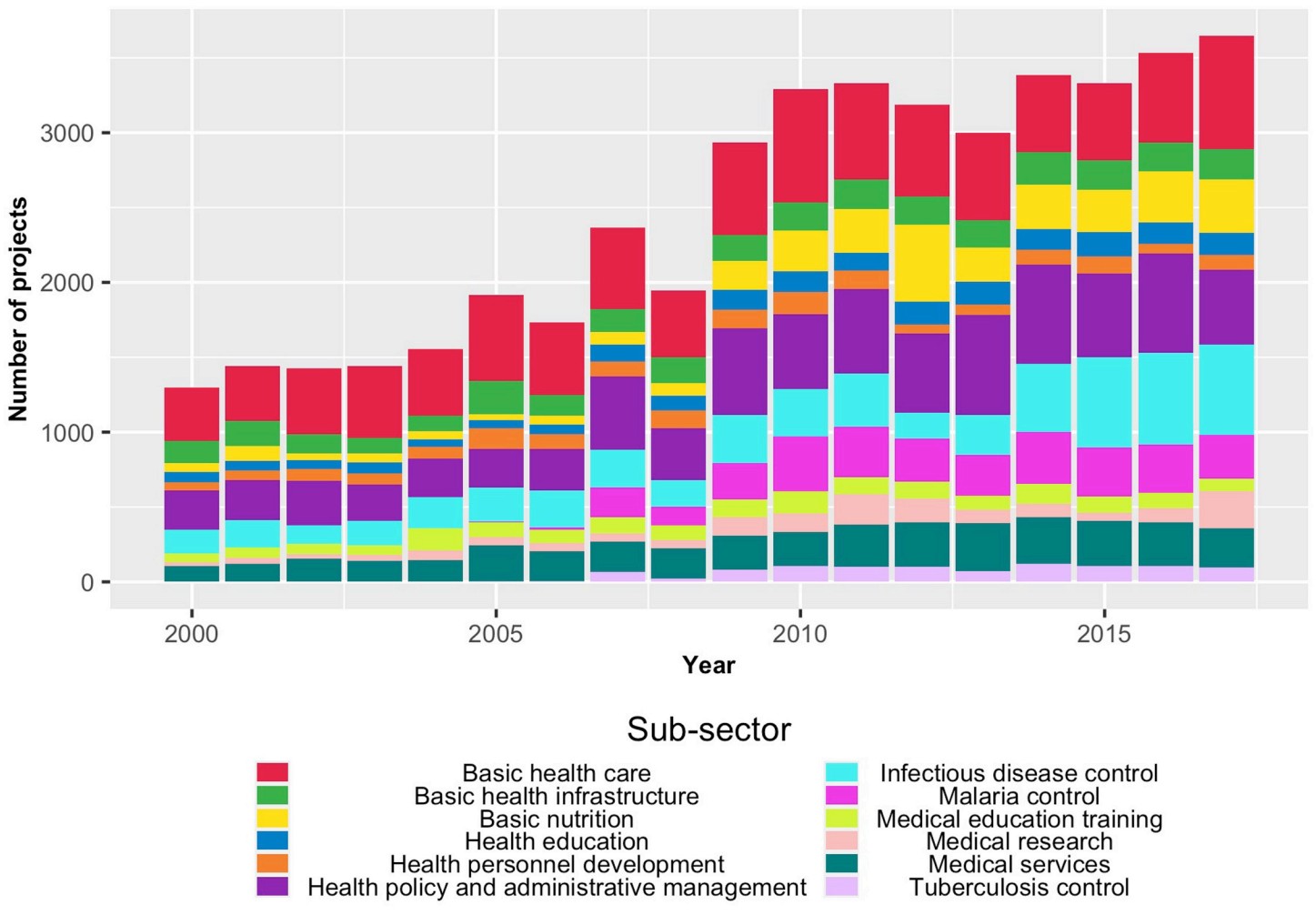

**Fig 4. Primary purpose of OECD health aid projects in Africa by year [2000–2017; N = 2016].**

hardest hit by the epidemic. During this period, Chinese healthcare teams provided prevention and treatment equipment such as protective suits, goggles, gloves, thermal scanners, and disinfectant sprayers. In addition, China purchased ambulances, built disease control centers, and dispatched public health experts for Ebola response activities [29]. China could be so responsive because it already had a presence of medical professionals on the ground, given its long-standing health engagements on the continent. China's active engagement and significant funding for Ebola-related prevention and care indicate its growing role as a leader within the global health community [29]. The language in the 2014 white paper that came out around the time of Ebola signals a move away from aid based on ideological reasons and instead towards aid to help tackle global security challenges, including health security [30]. Evidence of China's focus on improving people's lives indicates they are increasingly aware of the value of human security outlined and encouraged by the UN and the international community [2]. This does not mean that China has wholly abandoned non-interference as a fundamental foreign policy principle, but it demonstrates the value of cooperation [29].

Third, although our dataset ends in 2017, which predates the COVID-19 pandemic, China has supported COVID-19 efforts globally in various ways: supplies, medical teams, vaccines, infrastructure, multilateral support, and debt relief for indirect COVID effects. This support

was highlighted in China's 2021 foreign aid white paper that indicates COVID-19 is a priority and that China will prioritize global health for all in the coming years. In addition, the paper highlights China's intentions to prioritize public health systems in Africa and to uphold its commitments to achieve targets: Sustainable Development Goals [SDGs], China's Belt and Road Initiative [BRI], and the Forum on China-Africa Cooperation [FOCAC].

The methodology utilized to support the creation of the GCDF 2.0 data incorporates significant improvements over previous versions. Instead of relying on media sources to identify individual projects, the process begins by systematically reviewing thousands of official sources. These include original unredacted loan agreements, government reports from Chinese implementation agencies or recipient country departments, and press releases [1]. Official source retrieval is conducted country by country to track the full range of financial and in-kind transfers. A systematic search procedure supplements this in Factiva, which draws on 33,000 media sources in 28 languages. The new data also has an enhanced focus on project implementation. This focus involves using an improved data collection set and quality assurance protocols to identify implementation details [1]. Despite the significant re-engineering of the methodology to support the data generation, there are known limitations. For example, the data relies partially on unofficial information [such as media reports], lacks monetary amounts for about 59% of the health aid projects, and does not cover actual disbursement amounts. Furthermore, most projects with missing financial values are related to medical teams [code: 12201] and other in-kind transfers, which do not carry financial amounts. With benchmark data on the market pricing of medical visits, tests, and hospital stays and assumptions around the number of daily visits and tests per physician and medical equipment, these activities could be approximately priced out. This implies that readers should consider the reported financial values of China's health interventions in Africa as underestimates.

Our study also points to important directions for future research. Recent literature highlights the need for a rigorous analysis of China's health aid programs, including the quantity of aid, program decision criteria, and methods for determining impact [3, 9]. Additionally, few quantitative studies have evaluated potential motives for Chinese health aid other than the presence of natural resources. Zhao et al. highlight that most quantitative studies fail to account for the recipient country's development status and political interest, recommending further investigation [8]. Few comprehensive studies attempted to determine the relative importance of the most common theorized motivations. Finally, existing theories about Chinese health aid seem partly extracted from theories about development assistance. Yet, Guillon & Mathonnat observe significant differences between aid sectors, including that social aid is more responsive to the economic needs of recipient countries and is also influenced by foreign policy considerations [31]. Research could seek to build upon their findings.

Taken together, our findings observe the changes in China's health aid, starting with a disease eliminated in China, then moving towards global health security and health system strengthening, followed by shaping the governance mechanism. There seem to be opportunities for collaboration in the health sector and lessons to learn from China's health funding model. Through improved coordination and cooperation between Chinese and Western donors, we can, in theory, achieve improved public health outcomes, benefiting many deserving recipients across Africa.

## Acknowledgments

Valuable research assistance was provided by Audrey Crouch and Greatness Emmanuel-King. We would also like to thank the students in the Fall 2022 section of KINE 406 for their thoughtful editing of this work.

## Author Contributions

**Conceptualization:** Carrie B. Dolan, Ammar A. Malik.

**Data curation:** Ammar A. Malik, Sheng Zhang.

**Formal analysis:** Ammar A. Malik, Sheng Zhang.

**Investigation:** Carrie B. Dolan.

**Methodology:** Carrie B. Dolan, Ammar A. Malik.

**Project administration:** Carrie B. Dolan.

**Validation:** Carrie B. Dolan.

**Visualization:** Sheng Zhang.

**Writing – original draft:** Carrie B. Dolan, Ammar A. Malik, Kaci Kennedy McDade, Eli Svoboda, Julius N. Odhiambo.

**Writing – review & editing:** Carrie B. Dolan, Ammar A. Malik, Sheng Zhang, Wenhui Mao, Kaci Kennedy McDade, Eli Svoboda, Julius N. Odhiambo.

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
