## [Decision Letter · Decision Letter 0]

4 Apr 2023

PGPH-D-22-02102

Chinese Health Funding in Africa: The untold story

Dear Dr. Dolan,

Thank you for submitting your manuscript to PLOS Global Public Health. After careful consideration, we feel that it has merit but does not fully meet PLOS Global Public Health’s publication criteria as it currently stands. Therefore, we invite you to submit a revised version of the manuscript that addresses the points raised during the review process.

We look forward to receiving your revised manuscript.

Kind regards,

Genevieve Cecilia Aryeetey, Ph.D

Academic Editor

Journal Requirements:

2. Please provide separate figure files in .tif or .eps format only and remove any figures embedded in your manuscript file. Please also ensure that all files are under our size limit of 10MB.

3. Fig 2: please (a) provide a direct link to the base layer of the map (i.e., the country or region border shape) and ensure this is also included in the figure legend; and (b) provide a link to the terms of use / license information for the base layer image or shapefile. We cannot publish proprietary or copyrighted maps (e.g. Google Maps, Mapquest) and the terms of use for your map base layer must be compatible with our CC-BY 4.0 license. 

Additional Editor Comments (if provided):

Request the authors to address the comments from the reviewers

Reviewers' comments:

Reviewer's Responses to Questions

**Comments to the Author**

1. Does this manuscript meet PLOS Global Public Health’s publication criteria? Is the manuscript technically sound, and do the data support the conclusions? The manuscript must describe methodologically and ethically rigorous research with conclusions that are appropriately drawn based on the data presented.

Reviewer #1: Yes

Reviewer #2: Yes

2. Has the statistical analysis been performed appropriately and rigorously?

Reviewer #1: Yes

Reviewer #2: Yes

3. Have the authors made all data underlying the findings in their manuscript fully available (please refer to the Data Availability Statement at the start of the manuscript PDF file)?

Reviewer #1: Yes

Reviewer #2: Yes

4. Is the manuscript presented in an intelligible fashion and written in standard English?

Reviewer #1: Yes

Reviewer #2: Yes

5. Review Comments to the Author

Reviewer #1: Intro: Consider adding additional context, specifically- how does this $ amount of aid compare to other countries’ health aid? How does China’s health aid compare to other infrastructure projects? (e.g. dams, tunnels, etc)

Line 93: “China…. does not typically align with countries’ national and subnational health priorities” – consider giving an example/evidence either here or in the discussion.

Line 163-164: DAC codes – what does DAC stand for? Define ODA and OOF (need to include full name before abbreviation)

Line 143-156: Consider including single sentence about where GCDF pulls data from (there is a good explanation in the discussion section, but it leaves readers wondering for the entire paper where this data is sourced from). Also consider sentence on what type of organization AidData is (non-profit? Government-affiliated? Based in a single country?)

Line 163-164: Exclusion of OOF-like and vague – how does this change the results? (consider addressing in limitations section)

Line 167: What was the “detailed protocol” for reclassifying projects? It might be helpful to show an example or a flowchart of the protocol

Line 205-208: How were “resource-rich” countries defubed? By GDP? Was there a standardized metric? Recommend including their definition/method of determination of “resource-rich”

Line 219-220: Consider further defining “basic health care” and “medical services” projects? Vaccines, medicines, drugs and vertical programs like Malaria much more intuitive. How do basic health care and medical services differ from each other?

Line 226: How many projects (what percent) missing financial data? Why do some projects have commitments valued and some not? Are there any trends? How could this skew/bias data? (Consider further discussion limitations)

Line 240-241: Do not agree that increasing scale and reach of malaria projects would “indicate the feasibility of Chinese malaria control projects in Africa.” Please clarify/re-word.

Line 251-253: Sentence could be improved for clarity. Furthermore, please clarify who DAC members are.

Line 257-259: Please rewrite sentence for clarity

Line 331: Please write out SDG, BRI, FOCAC

Line 334: What kinds of “official sources”?

Line 351-352: Interesting comment about differences between aid sectors. Consider further supporting this statement with an example.

Figures –make sure Y axis labeled on all figures with appropriate title + units, also make sure all Figure titles are as clear/descriptive as possible (ex. Fig 4 title could be more specific about including 30 DAC countries)

- Figure 2 (Does Right Map reflect “Number of Chinese Aid Projects” or is it “Number of Chinese Health Aid Projects” – if it is the former it might help to further specify that it is “All Project Types” or if it is the latter, recommend including the word Health to clarify)

- Figure 3 – make sure y axis is completely labeled (% of what). Also, what is “Basic Care”?

- Figure 4 – label Y axis, clarify in text how are basic healthcare and medical services different? Why is TB separate from other infectious disease?

Reviewer #2: The article demonstrated evidence-based review and analysis. It actually met the publication criteria of PLOS GLOBAL PUBLIC HEALTH.

My observation however, is that, the authors would have presented the figures in color ink to bring out the significance of the points and inferences. Is it possible to achieve this before final publication? I wish to strongly suggest this be given a thought.

Others:

Results: Line 1 200-2017. Please, replace it with 2000-2017.

The same is applicable to number 63. Please, edit accordingly.

6. PLOS authors have the option to publish the peer review history of their article (what does this mean?). If published, this will include your full peer review and any attached files.

**Do you want your identity to be public for this peer review?** For information about this choice, including consent withdrawal, please see our Privacy Policy.

Reviewer #1: **Yes: **Haley Tupper

Reviewer #2: **Yes: **Khadijat Toyin Musah

---

## [Decision Letter · Decision Letter 1]

25 May 2023

Chinese Health Funding in Africa: The untold story

PGPH-D-22-02102R1

Dear Carrie Dolan,

We are pleased to inform you that your manuscript 'Chinese Health Funding in Africa: The untold story' has been provisionally accepted for publication in PLOS Global Public Health.

Best regards,

Genevieve Cecilia Aryeetey, Ph.D

Academic Editor

Reviewer Comments (if any, and for reference):

Reviewer's Responses to Questions

**Comments to the Author**

1. If the authors have adequately addressed your comments raised in a previous round of review and you feel that this manuscript is now acceptable for publication, you may indicate that here to bypass the “Comments to the Author” section, enter your conflict of interest statement in the “Confidential to Editor” section, and submit your "Accept" recommendation.

Reviewer #1: All comments have been addressed

2. Does this manuscript meet PLOS Global Public Health’s publication criteria? Is the manuscript technically sound, and do the data support the conclusions? The manuscript must describe methodologically and ethically rigorous research with conclusions that are appropriately drawn based on the data presented.

Reviewer #1: Yes

3. Has the statistical analysis been performed appropriately and rigorously?

Reviewer #1: Yes

4. Have the authors made all data underlying the findings in their manuscript fully available (please refer to the Data Availability Statement at the start of the manuscript PDF file)?

Reviewer #1: Yes

5. Is the manuscript presented in an intelligible fashion and written in standard English?

Reviewer #1: Yes

6. Review Comments to the Author

Reviewer #1: (No Response)

7. PLOS authors have the option to publish the peer review history of their article (what does this mean?). If published, this will include your full peer review and any attached files.

**Do you want your identity to be public for this peer review?** For information about this choice, including consent withdrawal, please see our Privacy Policy.

Reviewer #1: **Yes: **Haley Tupper
